

# Knowledge enhanced bottom-up affordance grounding for robotic interaction

Wen Qu, Xiao Li and Xiao Jin

Computer Science and Technology, Dalian Martime University, Dalian, Liaoning, China

## ABSTRACT

With the rapid advancement of robotics technology, an increasing number of researchers are exploring the use of natural language as a communication channel between humans and robots. In scenarios where language conditioned manipulation grounding, prevailing methods rely heavily on supervised multimodal deep learning. In this paradigm, robots assimilate knowledge from both language instructions and visual input. However, these approaches lack external knowledge for comprehending natural language instructions and are hindered by the substantial demand for a large amount of paired data, where vision and language are usually linked through manual annotation for the creation of realistic datasets. To address the above problems, we propose the knowledge enhanced bottom-up affordance grounding network (KBAG-Net), which enhances natural language understanding through external knowledge, improving accuracy in object grasping affordance segmentation. In addition, we introduce a semi-automatic data generation method aimed at facilitating the quick establishment of the language following manipulation grounding dataset. The experimental results on two standard dataset demonstrate that our method outperforms existing methods with the external knowledge. Specifically, our method outperforms the two-stage method by 12.98% and 1.22% of mIoU on the two dataset, respectively. For broader community engagement, we will make the semi-automatic data construction method publicly available at https://github.com/wmqu/Automated-Dataset-Construction4LGM.

## INTRODUCTION

Natural language is one of the most intuitive and flexible ways for humans to communicate with robots. Without the requirement of complex programming languages or graphical interfaces, natural language enables more natural and convenient interactions between humans and robots. Therefore, language-following robot manipulation has attracted increased attention.

Learning how to follow language instructions involves dealing with a symbolic grounding between the language instructions and robot perception and action, which is a challenging problem. Object affordance, as defined by *Gibson (1977)*, refers to the functional aspect an object part can provide. Understanding and grounding these affordances in natural

Corresponding author
Wen Qu, quwen@dlut.edu.cn

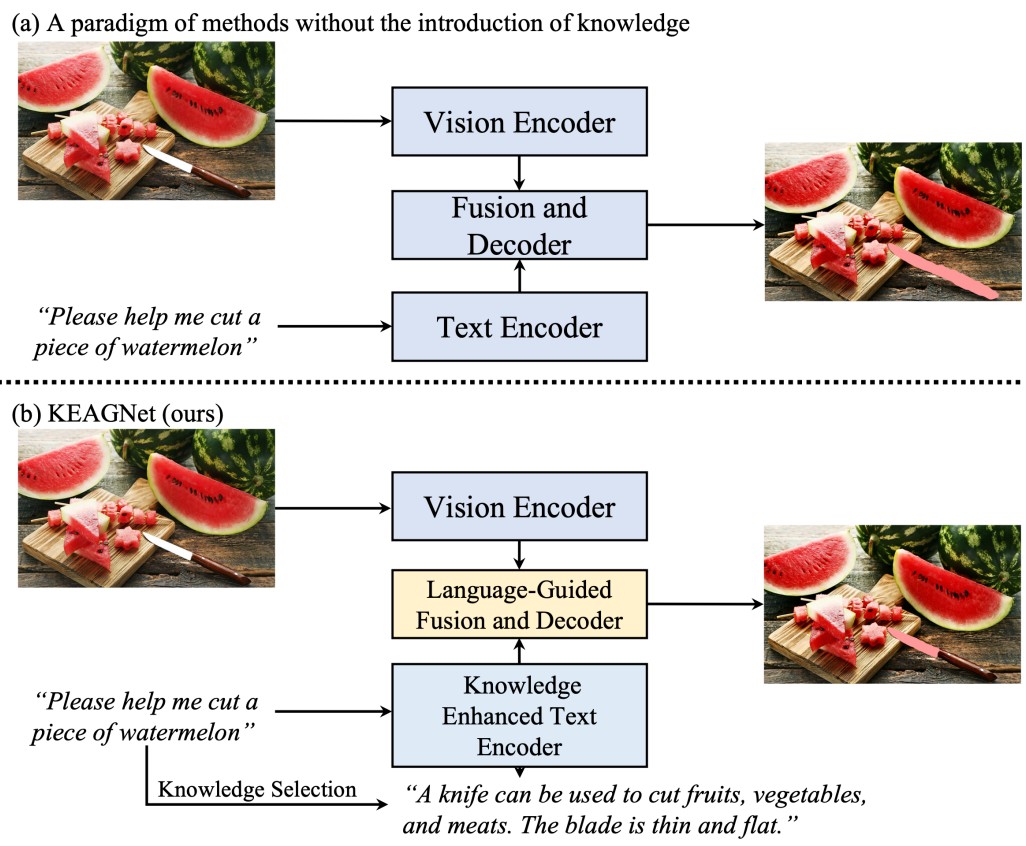

**Figure 1** **Object affordance segmentation tasks take an image and a text description as input, predicting masks for the specified objects mentioned in the description.** (A) Most methods employ visual-linguistic encoders and decoders to predict object-level masks. (B) We propose knowledge enhancement based on natural language instructions to form new text features, followed by utilizing visual-linguistic encoders and decoders to predict part-level masks. Image source credit: ID 114060752, © Ian Andreiev, Dreamstime.com.

language is a critical and intuitive means for agents to communicate effectively with humans. For example, consider a natural language instruction like "Help me cut this watermelon." Accompanied by an image, as shown in Fig. 1, humans can effortlessly identify the target object, "knife," and understand that the "blade" of the knife is the specific part meant for the task of "cutting".

In recent years, the mainstream methods adopt supervised deep learning methods and have shown promising results. However, in the domain of affordance grounding, two primary unresolved issues still persist: First, the task underscores the importance of commonsense knowledge in identifying the target object and its relevant affordance part. Most existing methodologies in affordance detection, such as those outlined in *Yin & Zhang (2022)*, *Zhang et al. (2022)* and *Chen et al. (2024)*, predominantly treat the task as an image semantic segmentation problem. These methods are typically limited to a set of predefined affordance categories and fail to capture the subtleties required for language-driven fine-grained segmentation. As depicted in Fig. 1A, previous studies have

often processed text and images through separate encoders, using a decoder subsequently to generate object-level segmentation masks. However, understanding affordances in some instances, such as recognizing that a 'knife is used to cut', necessitates commonsense knowledge. This aspect is often overlooked, thus neglecting the role of knowledge in affordance grounding. Consequently, the first key question arises: How can we more effectively integrate knowledge-enhanced text features with visual features to achieve more accurate, context-aware segmentation?

Additionally, deep learning approaches typically require offline datasets comprising language instructions and visual data (such as images or videos). Existing works *Hristov et al. (2017)*, *Ahn et al. (2018)*, *Hatori et al. (2018)*, *Magassouba et al. (2019)*, *Chen et al. (2020)*, *Mi et al. (2020a)*, *Shridhar, Mittal & Hsu (2020)*, *Mi et al. (2020b)*, *Nguyen et al. (2020)*, *Shridhar et al. (2020)*, *Huang et al. (2022)* have created various datasets under differing conditions for model training and testing. Although synthetic datasets are more accessible in a simulated environment, methods trained on synthetic data must address the challenge of domain shift problem when applied in a real-world scenarios. Realistic datasets, typically sourced from actual environments, such as MS COCO (*Lin et al., 2014*), ImageNet (*Deng et al., 2009*), or Kinect cameras, are used for language following target grounding, where instructions are manually generated and the most relevant objects in the visual data are annotated according to these instructions. However, annotating these realistic datasets tend to be a time-consuming and labor-intensive process. This leads to the second critical inquiry: How can we efficiently acquire a realistic, fine-granularity dataset for the language following manipulation grounding task?

To address the aforementioned issues and limitations, we propose the knowledge enhanced bottom-up language-guided affordance grounding network (KBAG-Net), which enhances visual language understanding through external knowledge. Our approach incorporates extra knowledge *via* a bimodal language feature interaction module, which then merges with visual features using a bottom-up fusion strategy. This method allows the enriched language features to guide low-level visual features during the process of affordance segmentation, ensuring a more context-aware output. Additionally, we propose a semi-automatic data generation method aimed at constructing a comprehensive language-following manipulation grounding benchmark from realistic images. This method automates the generation of data instructions, providing multiple types of annotations (both object-level and part-level) that correspond to these instructions within a unified framework, as shown in Fig. 2. By leveraging this approach, the manual effort required for dataset construction is substantially reduced, thereby enabling quicker generation of datasets tailored for diverse application scenarios.

In conclusion, grounding affordances in semantic representations facilitates communication and collaboration between humans and robots. By expressing affordances in a language-agnostic and interpretable manner, robots can effectively convey their intentions and reasoning to humans, fostering mutual understanding and trust. This work not only addresses the critical need for better integration of knowledge into deep learning models for affordance detection but also significantly alleviates the burdens associated with

**Multiple instruction types**

| Multiple annotation granularities | | Explicit instruction | | Implicit instruction |
|---|---|---|---|---|
| | | I want to use the pan on the bottom to cook. | Grasp the pan on the bottom. | Give me something that can cook. |
| | Object-level bounding box | | | |
| | Part-level segmentation mask | | | |

**Figure 2** **An example snippet of generated language following manipulation grounding dataset.** Multiple types refers to explicit instructions and implicit instructions. Multiple granularities refer to object-level and part-level annotation for images. Image source credits: *Nguyen et al. (2017)*.

the creation of realistic, finely-granulated datasets. The key contributions of this work are as follows.

- Knowledge Enhanced Affordance Grounding Method: We propose a bottom-up language-guided multimodal fusion network, which facilitates dense learning of the correlations between image feature information and knowledge-enhanced text feature information.
- Unified Framework for Dataset Generation: We have integrated three effective components into a unified framework and employed a pre-trained large language model to generate instructions. This unified approach significantly reduces the manual effort required for dataset construction and facilitates rapid dataset construction tailored to various application scenarios.
- The proposed method has been empirically validated to exhibit finer granularity and higher precision in object-level affordance segmentation compared to existing approaches.

In the subsequent sections, we begin by reviewing the related work in the field, and then detail the architecture and mechanisms of our proposed knowledge enhanced bottom-up language-guided affordance grounding network. Following the methodology, we introduce a unified framework designed for semi-automatic generation of datasets. Finally, we evaluate the performance and efficiency of both KBAG-Net and our dataset generation framework, followed by a conclusion of the work in the final section.

## RELATED WORKS

### Language conditioned manipulation grounding

Language conditioned manipulation grounding tasks have attracted extensive attention recently. In contrast to traditional visual perception tasks with predefined object category labels, the language conditioned manipulation grounding task involves more intricate language and visual information. Some works *Chen et al. (2020)*, *Mi et al. (2020a)*, *Nguyen et al. (2020)*, *Shridhar et al. (2020)* focus on grounding tasks for different objects, while some works *Hristov et al. (2017)*, *Ahn et al. (2018)*, *Magassouba et al. (2019)*, *Hatori et al., (2018)*, *Shridhar, Mittal & Hsu (2020)*, *Mi et al. (2020b)* focus on the grounding of spatial relationships and attributes of objects. Several studies have focused on human–robot interactions with natural language. Utilizing question asking can increase accuracy and eliminate ambiguity (*Hatori et al., 2018*; *Chen et al., 2020*). In order to improve the generalization ability, some end-to-end approaches attempt to scale and broaden the data collected (*Mees, Hermann & Burgard, 2022*; *Jang et al., 2022*). According to the data types, the existing datasets in Table 1 can be divided into simulated dataset and realistic dataset. The simulated datasets are primarily employed for tasks that involve extended interactions based on natural language instructions (*Shridhar et al., 2020*; *Mees, Borja-Diaz & Burgard, 2023*), which focus on learning robotic control directly from image-instruction pairs. However, the simulated datasets have the problem of domain shifting in practical applications. In addition, most realistic datasets are manually annotated, resulting in higher costs. Automatically constructing datasets faces the challenge of lacking diverse in instructions and fine-granularity annotations for images. To address these challenges, we propose a unified framework that generates language instructions and fine-granularity image annotations simultaneously.

### Object affordance detection

The concept of affordance consists of object affordance and environment affordance. In this article, we specifically focus on the visual affordance of objects in images. *Myers et al. (2015)* propose jointly applying geometric properties and local shapes to identify object affordance. *Do, Nguyen & Reid (2018)* propose AffordanceNet to simultaneously detect the affordance of multiple objects in RGB images. More recently, *Mi et al. (2020a)* propose an attention-based architecture to learn the affordance of objects. *Zhai et al. (2022)* propose OSAD-Net for detecting object affordances in a scene through collaboration learning. *Zhao, Cao & Kang (2020)* utilize a relationship-aware network to implement affordance segmentation in an end-to-end way. In addition, several researchers have introduced context like the object shape (*Chen et al., 2024*), boundary (*Yin & Zhang, 2022*) to improve the performance of affordance segmentation. On one hand, these methods neglect the correlation between natural language and object affordance. In our approach, we segment the corresponding affordance part, which better aligns with realistic scenarios. On the other hand, existing methods lack of the commonsense for affordance learning. In this work, we introduce external knowledge to understand the usability of objects better.

**Table 1   List of typical natural language manipulation grounding datasets.** The blank space in the table indicates that it is not detailed in the article.

| Work | Natural language | Object and task categories | Language generation method | Statistics of corpus |
|---|---|---|---|---|
| *Hristov et al. (2017)* | Pick up the blue cube. Put the red block on top of the yellow cell. | 3 for shape (cell, block, cube); pick up, put, drop, take | | 2,000 labeled image patches, 4,000 symbols |
| *Ahn et al. (2018)* | Pick up the yellow thing on the right column. | blocks; pick up | | 477 images, 20,349 unambiguous language commands, 7,119 ambiguous language commands |
| *Hatori et al. (2018)* | Pick the white packet in center and put it into the upper left box. | commodities and daily household items; pick and place | human-annotated on Amazon Mechanical Turk | 1,180 images, 91,590 language instructions |
| *Magassouba et al. (2019)* | Bring me the empty bottle from the right wooden table. | fetch | annotated by an expert user | 308 images, annotate 1,010 targets |
| | Move the object with the black lid to the top left box. | commodities and daily household items; pick-and-place | human-annotated on Amazon Mechanical Turk | 1,180 images, 91,590 language instructions |
| *Chen et al. (2020)* | Could you please pour me some water? | blend, pour, fry, brush, dip, dump, fill, heat, rub, sprinkle, season | | |
| *Mi et al. (2020a)* | I am thirsty, I want to drink some water. | 42 objects are commonly used in household; object affordance detection | human-annotated | 12,349 RGB images and 14,695 bounding box annotations for object affordance detection |
| *Shridhar, Mittal & Hsu (2020)* | Pick up the toy with the black and yellow face. Pick up the leftmost blue cup. | pick up and place | human-annotated | 3,000 images, 21,586 instructions |
| *Mi et al. (2020b)* | Bring me the red cup in the box and the second bottle from the left. | pick up, move | human-annotated | 163 images (133 from RefCOOCO, 30 captured by Kinect V2), 415 expressions |
| *Nguyen et al. (2020)* | Give me an item that can contain. Hand me something to eat. | object affordance detection | template | 655 verb-object pairs over 50 verbs and 216 object classes |
| *Shridhar et al. (2020)* | Rinse off a mug and place it in the coffee maker. | long horizon tasks | | 25,743 language directives corresponding to 8,055 expert demonstration episodes |
| *Huang et al. (2022)* | A bottle is in the right back of the book. Put the bottle to the right side of the book. | 100 kinds of daily objects; pick and place | template | PD dataset: 41 expressions for 8 kinds of relative spatial relations and 14 kinds of adjectives; CA dataset: 47 expressions for 10 kinds of orientation relations |

**Table 1** (_continued_)

| Work | Natural language | Object and task categories | Language generation method | Statistics of corpus |
|------|------------------|---------------------------|---------------------------|---------------------|
| _Mees et al. (2022)_ | "open drawer" → "push block in drawer" → "pick object from drawer" → "stack blocks" → "close drawer" | blocks, light bulb, LED; long horizon tasks, _e.g._, Rotate, Push, Move slider, Open/close, Lift | collect from crowd-sourced natural language instructions | over 400 crowd-sourced natural language instructions corresponding to over 34 tasks and label episodes |

### Prompt learning

Prompt learning, a recent trend in the field of few-shot learning of NLP (_Jiang et al., 2020_; _Brown et al., 2020_; _Zhu et al., 2023_), involves leveraging pretrained language models through cloze-style prompts to enhance the performance of downstream tasks. _Brown et al. (2020)_ propose a method for adjusting the behavior of a frozen GPT-3 model through prompt design. In order to efficiently deploy prompt-learning pipelines for researchers and developers, _Ding et al. (2021)_ present OpenPrompt that modularizes the entire framework of prompt-learning. In the latest research, prompt learning is applied to large vision-language models in computer vision (_Du et al., 2022_; _Zhou et al., 2022b_; _Zhou et al., 2022a_). CoOp (_Zhou et al., 2022b_) turns discrete words in a prompt into continuous prompt learning for adapting pre-trained vision-language models. In order to generalize to unseen classes, CoCoOp (_Zhou et al., 2022a_) generates an input-conditional prompt instead of the fixed learned vectors for each input image.

In general, to determine the verb associated with an object, the common approach is to establish a mapping between the object's category and its corresponding verb. However, the method of dictionary mapping is a challenge for unseen objects to generate corresponding verbs. To address this issue, we utilize a large language model as a knowledge base and use prompt-learning to predict the probability distribution across the vocabulary for the <MASK> token's position, thus obtaining the verb of the object.

## METHODOLOGY

In this section, we first present the overall framework of our approach. As shown in Fig. 3, image $I$ and expression $X$ are input into a multi-modal encoder for feature extraction respectively. The language instructions undergo enhanced encoding through external knowledge $K$, and the visual encoder concurrently extracts low-level and high-level representations of visual features. Subsequently, the bimodal language feature interaction module refines the text features that have been enhanced with knowledge. Following this, the enhanced text features are fused separately with the low-level and high-level visual features. Ultimately, the amalgamated multi-modal features are fed into the part-level affordance grounding module to achieve the affordance segmentation of objects, yielding the final segmentation mask $f_{mask}$.

### Multimodel encoder

**Visual encoder.** Given an image $I \in R^{H \times W \times 3}$ with the height $H$, the width $W$ and three color channels, we employ ResNet101 (_He et al., 2016_) as the visual backbone, a deep

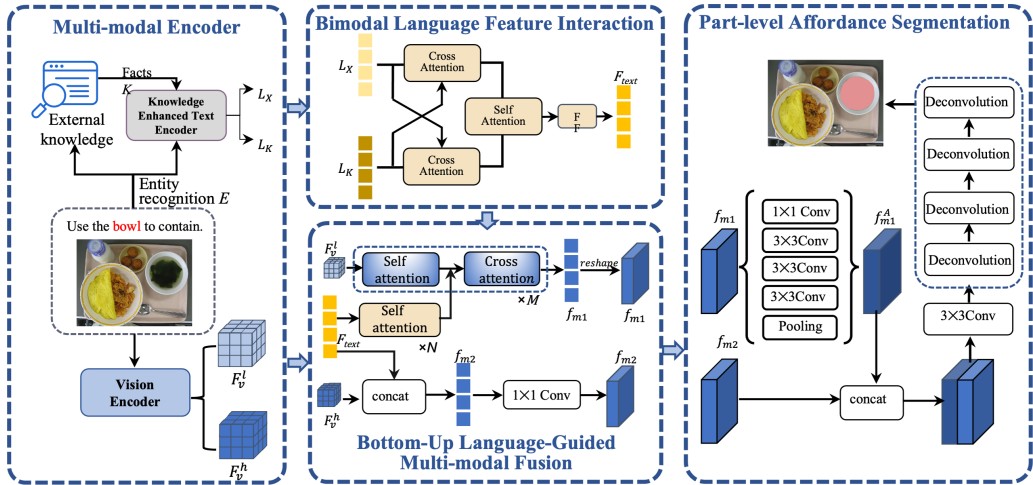

**Figure 3** An overview of knowledge enhanced bottom-up language-guided affordance grounding network (KBAG-Net). Image source credits: *Nguyen et al. (2017)*.

residual network widely recognized for its effectiveness in various computer vision tasks. In our model, we denote $res\_i$ as the output features of the $i-th$ block in ResNet101. Specifically, we use $res\_2$ and $res\_5$ to represent low-level visual features $F_v^l$ and high-level visual features $F_v^h$, respectively. The choice of these specific layers is strategic: low-level features are adept at capturing intricate details such as edges and textures, which are essential for detailed image analysis. On the other hand, high-level features excel at capturing advanced semantic information about objects, providing a broader context and understanding of the objects in the images. Our approach aims to fully leverage the strengths of both low-level and high-level visual features.

**Knowledge enhanced text encoder.** In our approach, we handle an expression $X = \{x_1, \ldots, x_m\}$ consisting of $m$ words, where $x_i$ represents the $i-th$ word in the sequence. Our initial step involves utilizing an entity recognition algorithm to identify and extract object entities within the expression, which we denoted the entities as $E = \{e_1, \ldots, e_n\}$ with $n < m$. Each entity serves as a key to retrieve corresponding knowledge from the collected knowledge. The collection of knowledge will be illustrated in the experiments section.

For every identified entity $e_j \in E$ in the expression $X$, we match it with a corresponding fact $k_j$ in our knowledge base. Each fact $k_j$ is composed of multiple sentences, providing a rich context for the entity. To process and extract features from both the expression $X$ and the knowledge $K$, we employ BERT (*Devlin et al., 2019*), an effective language representation model to capture contextual features. The features extracted from the expression and the knowledge are denoted as $L_X$ and $L_K$, respectively. Then, we design a bimodal language feature interaction module to align and fuse the two features. This process of knowledge selection is pivotal, as it significantly enhances the model's capability to comprehend and interpret natural language instructions with greater semantic context.

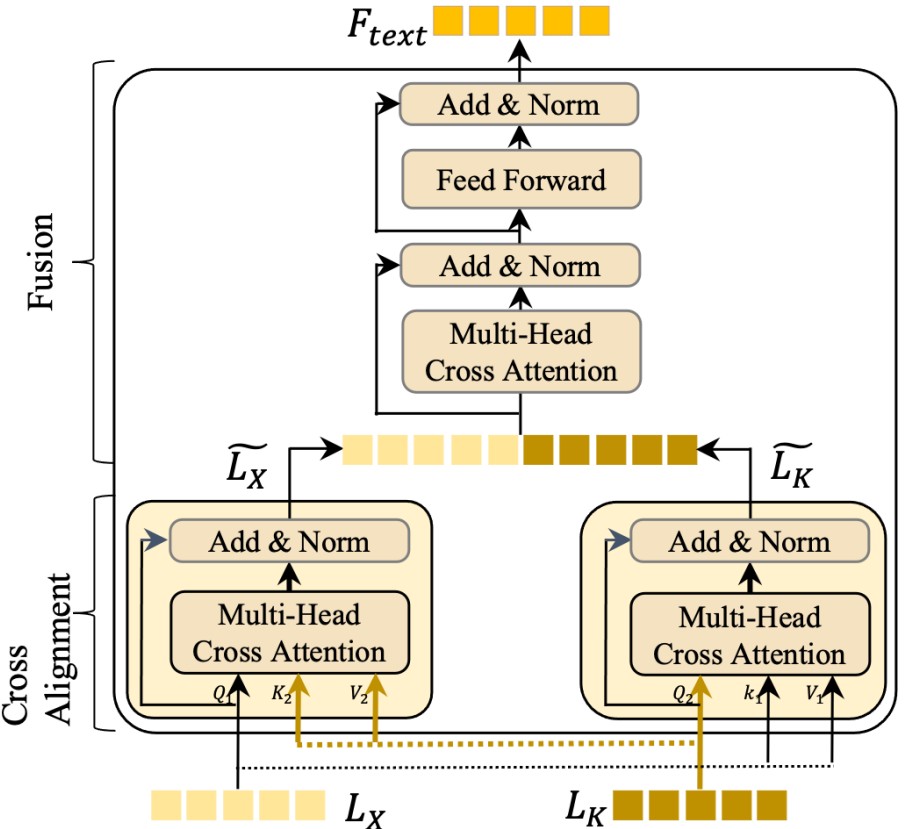

**Figure 4** **An illustration of the bimodal language feature interaction (BLFI) module.**

## Bimodal language feature interaction

The bimodal language feature interaction (BLFI) module is a cornerstone of our approach, designed to enhance the understanding of instructions by focusing on two key aspects: (a) emphasizing key information within $L_X$ and $L_K$, and (b) filtering out irrelevant information from these features. This dual focus ensures that our model captures the most significant elements of the language and knowledge features. As depicted in Fig. 4, the BLFI module operates in two distinct stages: knowledge alignment and knowledge fusion.

### Knowledge alignment

In this initial stage, we address the semantic discrepancies between $L_x$ (language features) and $L_K$ (knowledge features) by implementing two cross-attention modules. In the cross-attention mechanism, one type of feature acts as the query, while the other serves as both the key ($K$) and value ($V$).

Weight matrices $W_j^Q$, $W_j^K$, and $W_j^V$ are used to update the corresponding $K$, $Q$, and $V$, as shown in Eqs. (1) and (2). For instance, $L_X$ functions as $Q_1$, with $L_K$ being $K_1$ and $V_1$. The attention mechanism then computes the attention values, leading to a more aligned feature representation with the following equation.

$$MH(L_X, L_K, L_K) = Concat(head_1, head_2, \ldots, head_h)W^0, \tag{1}$$

$$head_j = Attention(L_X W_j^Q, L_K W_j^K, L_K W_j^V), \tag{2}$$

$$\tilde{L_X} = Attention(Q, K, V) = \sum \frac{1}{z} exp(\frac{QK^T}{\sqrt{d_k}})V. \tag{3}$$

In Eq. (1), $W^0$ denotes the learnable matrix. $z$ in Eq. (3) represents the normalization factor and $V$ are knowledge features. $d_k$ denotes the dimension of features. The computation of $\tilde{L_K}$ is similar to that of the $\tilde{L_X}$.

### Knowledge fusion

Following the alignment stage, $L_X$ undergoes layer normalization and is connected *via* residual connections, resulting in an enhanced feature set $\tilde{L_X}$. A similar process is applied to $L_K$, yielding $\tilde{L_K}$. Although space constraints limit a detailed description, it is essential to note that this stage is pivotal for merging the aligned features, thereby creating a unified and enriched representation that encapsulates both the textual and knowledge-based aspects of the data.

Through these stages, the BLFI module ensures that the model not only identifies but also emphasizes the crucial elements in the language and knowledge features, effectively filtering out less relevant information. This refined processing enhances the model's ability to understand and interpret instructions, leading to more accurate and contextually rich outputs.

## Bottom-up language-guided multi-modal fusion

Our model employs a bottom-up language-guided multi-model fusion (BULG) module, an innovative approach to integrative visual features with knowledge-enhanced text features. Traditional multimodal methods often rely on attention mechanisms to fuse text and visual features. These methods, as illustrated in Figs. 5A, 5B, 5C, typically focus on object-level alignment using text features that contain limited information. However, in our approach, the knowledge-enhanced text features are rich in semantic content, playing a pivotal role in part-level affordance grounding.

Inspired by structure (c) from Fig. 5 and the work on attention mechanism (*Vaswani et al., 2017*), our novel BULG structure addresses two critical factors. (1) Enhanced role of text information: We employ multi-layer self-attention mechanisms to iteratively extract information from text features. This method allows for a deeper understanding of the text content, leveraging its rich semantic nature. (2) Fine-grained multimodal alignment: Given the details characteristic of the task, our model requires precise alignments between text and visual features. This alignment is crucial for accurately capturing the nuanced interplay between different modalities.

In the BULG module, illustrated in Fig. 6, we combine a intra-modal interaction module and a cross-modal interaction module. The right part of the module takes text features $F_{text}$ as input, employing $N$ layers of self-attention to generate refined text features $f_{text}$. Simultaneously, in the left part, the input comprises $f_{text}$ and low-level visual features $F_v^l$.

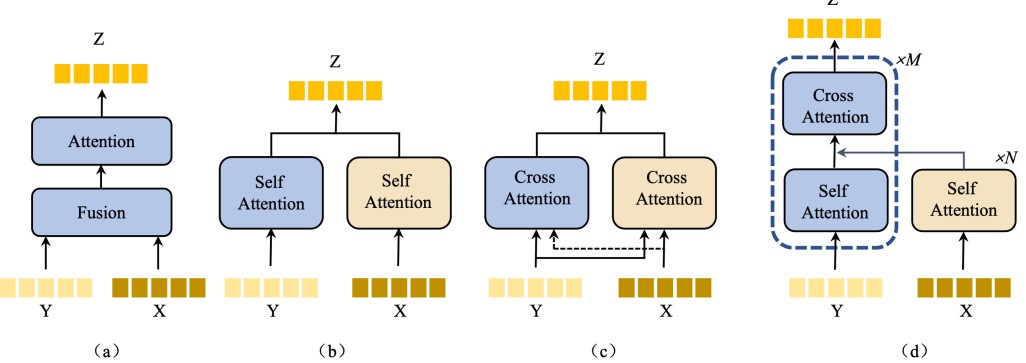

**Figure 5** An illustration of the structure of existing attention-based multimodal fusion methods.

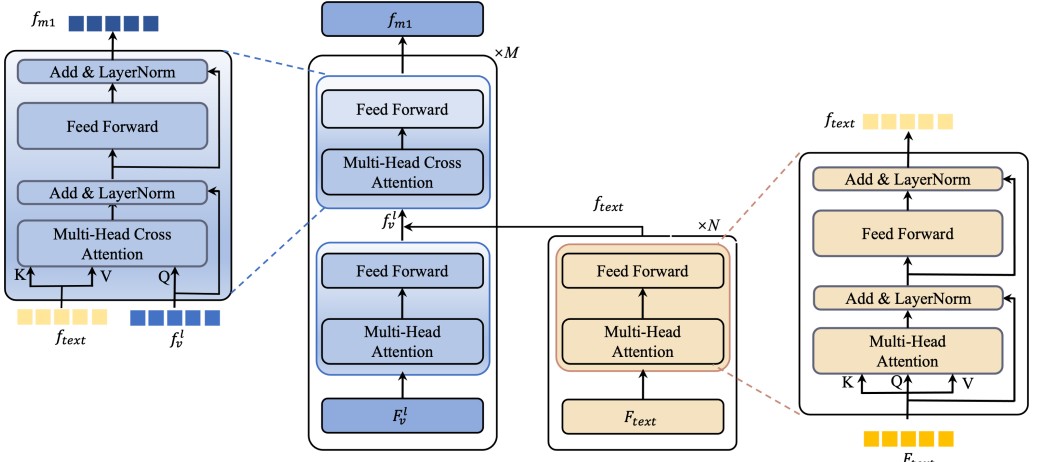

**Figure 6** An illustration of the bottom-up language-guided multi-modal (BULG) module.

Initially, $F_v^l$ undergoes unimodal interaction to extract critical visual features from $f_v^l$. These features are then combined with $f_{text}$ in the multimodal interaction module, employing a bottom-up guided attention mechanism to enhance visual features learning. This process results in the generation of multimodal features $f_{m1}$.

For the fusion process, we handle text and visual features distinctly based on their level. With low-level visual features $F_v^l$, we apply a bottom-up text-guided approach to form $f_{m1}$. High-level visual features $F_v^h$, containing rich object-level semantic information, are directly fused with text features to produce multimodal features $f_{m2}$. This process is described mathematically as:

$$f_{m2} = Concat\left(g\left(F_v^h \cdot W_v\right) \cdot g\left(F_{text} \cdot W_t\right)\right). \tag{4}$$

Here, $W_v$ and $W_t$ are two transformation matrices that align the visual and text representations into a uniform feature dimension, facilitating effective fusion. 'Concat' is the concatenation operation. $g$ denotes the LeakyRelu activation function.

### Part-level affordance grounding

In this subsection, we detail how our model leverages text-guided multimodal features to detect various affordances in images.

Our process begins with the integration of multimodal feature $f_{m1}$ and $f_{m2}$. $f_{m1}$ is the multi-model feature from bottom-up language-guided multi-modal block. The first step involves passing $f_{m1}$ through an ASPP module, as described in the DeepLab framework by *Chen et al. (2017)*. ASPP is effective in capturing multi-scale contextual information, which is crucial for affordance understanding in complex images. The output feature from ASPP is denoted as $f_{m1}^A$ and is calculated as follows:

$$f_{m1}^A = ASPP(f_{m1}). \tag{5}$$

Next, we concatenate $f_{m1}^A$ with $f_{m2}$ and pass this combined feature through our segmentation module. This process yields the final segmentation result, represented as:

$$f_{mask} = unsample(Concat(unsample(f_{m1}^A), f_{m2})). \tag{6}$$

By employing filters with various sampling rates and fields of view, ASPP allows the model to capture both detailed and broader contextual elements in images. To achieve more precise segmentation results, we upsample the feature map by a factor of four, ensuring a finer granularity in the final output.

### Loss function

For training our model, we utilize the Sigmoid Binary Cross Entropy (BCE) (*Mao, Mohri & Zhong, 2023*) loss function. This loss function is particularly suited for binary classification tasks, such as segmenting specific affordances in images. The BCE loss for our task is defined as:

$$L = \sum_{l=1}^{H \times W} [y_l log(p_l) + (1 - y_l) log(1 - p_l)], \tag{7}$$

where $y_l$ and $p_l$ are the $l$-th elements of the ground-truth mask and predicted mask $f_{mask}$, respectively. Here, $W$ and $H$ denote the width and height of the image respectively.

## SEMI-AUTOMATIC DATASET CONSTRUCTION

In this section, we first introduce the notation and formulation used for semi-automatic dataset construction, and then describe the three components of the process in detail.

### Notation and formulation

Given an unlabeled image $I$, our goal is to automatically generate the annotation $A$ for the image. Specifically, each $A = \{L, B, M\}$ comprises a language instruction $L$ referring

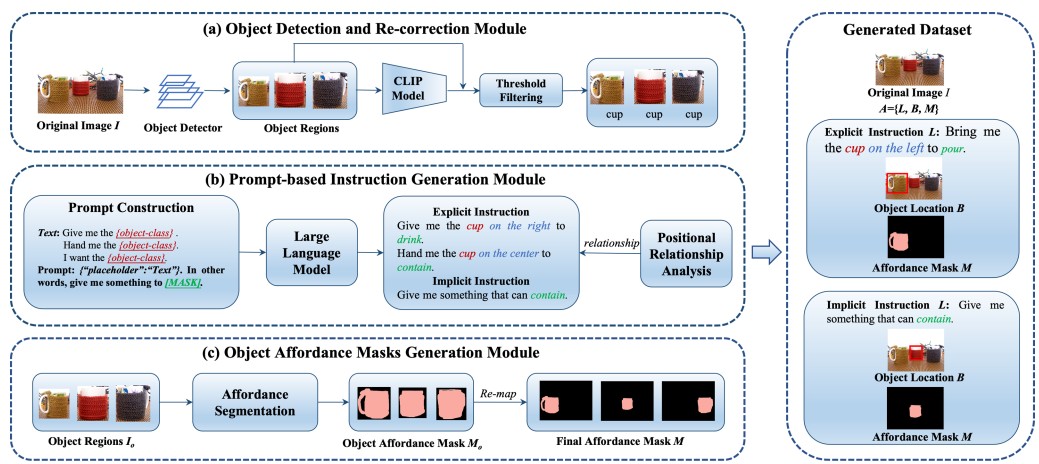

**Figure 7** **The framework of semi-automated dataset construction.** It consists of object detection and re-correction module, prompt-based instruction generation module and object affordance masks generation module. Image source credits: *Nguyen et al. (2017)*.

to a candidate object, the corresponding object location *B* and the affordance mask *M* of the object part. The affordance mask *M* is a mask of the object part that is related to the manipulation verb in the instruction, such as "contain" and "cut". Importantly, a candidate object may have both bounding box and mask annotations within our dataset generation process. The overall framework, as illustrated in Fig. 7, comprises three key components, each contributing to the automatic annotation process:

First, the object detection and re-correction module takes an unlabeled image as input, utilizes an object detector to predict the bounding box coordinate, and object regions accordingly. The cropped regions are then fed into the CLIP (*Radford et al., 2021*), which generates possible class labels for the cropped regions. The results of the object detector and the CLIP are compared to filter the wrong predictions. Then, the prompt-based instruction generation module automatically generates positional relationships and predicts verbs related to the candidate objects. Finally, the object affordance mask generation module is responsible for producing affordance masks for target objects using DeepLabV3+ (*Chen et al., 2018*). We provide detailed implementation insights in the following subsections.

## Object detection and re-correction module

In this subsection, we delve into the object detection and re-correction module, a critical component of our automatic data annotation process. Its primary objective is to enhance the quality of object class labels obtained from an initial object detector, such as Faster RCNN, and to mitigate issues like incorrect category predictions and the presence of redundant bounding boxes. To achieve this, we leverage the power of the Visual and Language model CLIP (*Radford et al., 2021*), which has demonstrated state-of-the-art capabilities in understanding visual concepts from a vast corpus of image-text pairs. CLIP comprises two encoders, one for text and another for images. We employ a specific input template, "This is a photo of a {label}", for the text encoder, where {label} represents the

object category outputted by the object detector. For the image encoder, we use the cropped region as input. These two encoders result in feature vectors $f_t$ and $f_v$, respectively. At last, the CLIP adopts a dot product to compute the similarities between two features ($f_t$ and $f_v$) from each modality, followed by Softmax-based probability predictions for all object categories.

Let $p_i$ denote the probability of a region proposal being predicted as the $i$th object category by CLIP, which is computed as Eq. (8). In the equation, $cos(\cdot, \cdot)$ denotes cosine similarity and $\tau$ is a temperature parameter learned by CLIP during training. $\max(p_i)$ is the maximum of the prediction possibilities.

$$p_i = \frac{\exp\left(\frac{\cos(f_t, f_v)}{\tau}\right)}{\sum_{j=1}^{K} \exp\left(\frac{\cos(f_t, f_v)}{\tau}\right)}. \tag{8}$$

When the object detector and CLIP both predict the object as the $i$th category, we first compute the average score $S_i$ of the object detector and the CLIP follow Eq. (9):

$$S_i = \frac{r_i + \max(p_i)}{2}. \tag{9}$$

To enhance the quality of category labels, we introduce a score $S_i$, computed as the average of the probability score from the object detector ($r_i$) and the maximum CLIP prediction probability ($\max(p_i)$), as expressed in Eq. (9). Then, the average score $S_i$ is compared with a score threshold $S_T$. This threshold filtering helps eliminate predictions with low confidence, ensuring that only high-confidence category predictions are considered. Additionally, we address cases where the object detector and CLIP provide conflicting category predictions. In such situations, we prioritize the prediction from CLIP to improve label accuracy. Predictions from the object detector that do not align with CLIP predictions are discarded, further refining the category labels. Figure 7A provides a visual representation of the process.

**Prompt-based instruction generation module**

This module is responsible for generating manipulation instructions, which can be categorized into two types: explicit and implicit instructions. The explicit instructions contain specific details, including the object category, verb, and positional relationship, providing comprehensive guidance for manipulation tasks. For example, "Hand me the cup on the right to contain". The implicit instructions focus primarily on the manipulation verb itself, omitting specific object category references. An example of an implicit instruction is "Hand me something to contain", where the object category is not explicitly mentioned.

Generating explicit instructions requires careful consideration of both positional relationships and appropriate verbs to convey the desired manipulation task. To achieve this, we employ a multi-step approach, as shown in Fig. 7B:

**Positional relationship analysis**: Distinguishing among multiple objects of the same class is challenging. We address this issue by analyzing spatial relationships in both horizontal (left, middle and right) and vertical (top and bottom) dimensions. By calculating the center coordinates of bounding boxes obtained from the object detection and re-correction module, the position relationship is determined by comparing the center

**Table 2  Templates for the *Text* and implicit instructions.**

| *Text* templates | Implicit templates |
|---|---|
| I need the $\{x_{inp}\}$ | An item that can {verb} |
| Hand me the $\{x_{inp}\}$ | An object that can {verb} |
| Pass me the $\{x_{inp}\}$ | Give me something that can {verb} |
| I want use the $\{x_{inp}\}$ | Give me an item that can {verb} |
| Bring me the $\{x_{inp}\}$ | Hand me something to {verb} |
| I want use the $\{x_{inp}\}$ | Give me something to {verb} |
| Get the $\{x_{inp}\}$ | I want something to {verb} |
| Give me the $\{x_{inp}\}$ | I need something to {verb} |
| Fetch the $\{x_{inp}\}$ | |
| Bring the $\{x_{inp}\}$ | |

coordinates. These offsets help us construct instructions such as "Bring the cup on the left to contain" and "I want the bowl on the top to contain".

**Prompt construction**: Prompt-learning uses pre-trained language models as knowledge bases resulting in great performance on different downstream tasks. Using prompt-learning can automate the process of finding the object-verb pair relationship. Denote $x_{inp}$ as the original input and $x_{prompt}$ as the prompt that is fed into the MLM. The mapping from $x_{inp}$ to $x_{prompt}$ is performed using a template $m$. This template defines where each input sequence and the placement will be placed in the prompt. The template used in our work is "*{"placeholder": Text}. In other words, give me something to* [*MASK*]. ". The *Text* contains $x_{inp}$ as shown in Table 2. Feeding the prompt into the MLM will produce a probability distribution $p([MASK]|x_{prompt})$, which predicts the token most likely filled in the blank. The $x_{inp}$ is the output of the object detection and re-correction module. We use GPT-2 (*Radford et al., 2019*) as a pre-training language model and adopt OpenPrompt (*Ding et al., 2021*), an open-source and unified easy-to-use toolkit, to implement prompt-learning.

**Prompt-based instructions generation**: To ensure diversity in our instructions, we generate explicit and implicit instructions simultaneously. As shown in Table 2, we start by randomly selecting an explicit template and construct a *Text* from $x_{inp}$. The *Text* is then mapped to $x_{prompt}$, which is used as input to the MLM. The MLM outputs a probability distribution $p([MASK]|x_{prompt})$ describing which verb most likely fills in the mask. The verb with the highest probability will be added to the end of the explicit instruction and also serves as the basis for the implicit instruction.

## Object affordance masks generation module

After object detection and prompt-based instruction generation, we take a crucial step towards enhancing the realism and applicability of our generated dataset. Existing datasets typically provide object bounding boxes for each instruction, which may not suffice for tasks requiring part-level instruction grounding. Consider an instruction such as "Use the cup to contain". In such cases, the robot agent needs to focus on the specific part of the object that provides the "contain" affordance. To address this need, our goal is to provide not only the object bounding box but also the affordance mask for each instruction. This

addition enhances the practicality of the generated dataset and its suitability for a wider range of real-world applications. Our approach involves the following key steps, as shown in Fig. 7C.

**Object region cropping**: For each instruction, we begin by cropping the object region $I_o$ from the original image. This isolated region serves as the basis for affordance mask prediction.

**Affordance mask prediction**: We employ the DeepLabV3+ (*Chen et al., 2018*) segmentation model, fine-tuned on an affordance dataset, to predict the affordance category for each pixel within the object region. This prediction results in an object affordance mask $M_o$. Importantly, since an object may possess multiple affordance classes, the pixel values in the affordance mask $M_o$ are assigned as $n$ unique labels $\{0, 1, \ldots, n-1\}$. If the pixel values correspond to the affordance class specified in the instruction, they are kept and set as ones. Otherwise, they are set to zeros.

**Remapping of the affordance mask**: To align the object affordance mask $M_o$ with the coordinates of the object region, we remapped it to an initial affordance mask that is initialized as zeros and shares the same size as the original image. The outcome of this process is the pixel-wise segmentation of the object part $M$ corresponding to the instruction. This segmentation provides crucial information about which part of the object is relevant to the instruction, facilitating precise instruction grounding.

## EXPERIMENT

We designed our experimental setup to address two distinct yet related challenges: affordance grounding and semi-automatic dataset generation. The experiments were structured to evaluate the effectiveness and efficiency of our proposed methods through a series of tests and comparisons.

For affordance grounding challenge, we employed both comparison and ablation studies to assess the performance of KBAG-Net. The comparison study evaluates KBAG-Net against existing methods to establish its relative effectiveness. The ablation study, on the other hand, aims to isolate and understand the impact of each design choice within KBAG-Net on its overall performance.

Regarding the semi-automatic dataset generation framework, we utilized the framework to generate a dataset, which was then compared against a human-annotated dataset to evaluate the quality of the auto-generated dataset. This comparison aims to determine the effectiveness of the framework in reducing the labor-intensive process of manual dataset annotation. In summary, the experiments are specifically designed to address the following key research questions:

RQ1: How does KBAG-Net perform compared with existing related methods?

RQ2: How does each design choice in KBAG-Net affect its performance?

RQ3: How is the quality of generated data by our method compared with manual annotation?

RQ4: Does the proposed semi-automatic data construction method alleviate labor-intensive annotation?

## Experiments setup for affordance grounding

Before evaluating the semi-automatic data construction, it is crucial to first validate the effectiveness of KBAG-Net using manually annotated datasets. This initial step ensures that the performance measurements are not adversely influenced by any noise potentially introduced in the semi-automatically generated data. For this purpose, we utilized two well-established datasets: IIT-AFF VL and UMD VL (*Qu et al., 2024*). We first introduce the datasets in details, then present the evaluation metrics for the affordance grounding task. At last, the knowledge collection process is illustrated.

**Datasets** For a comprehensive evaluation of our method, we utilized two visual language datasets IIT-AFF VL and UMD VL dataset (https://github.com/WenQu-NEU/Affordance-Grounding-Dataset). The IIT-AFF VL and UM VL datasets comprises a total of 15, 905 language instruction expressions, each paired with a corresponding RGB image (images within original IIT-AFF (*Nguyen et al., 2017*) (https://sites.google.com/site/iitaffdataset/) and UMD (*Myers et al., 2015*) (https://users.umiacs.umd.edu/~fer/affordance/part-affordance-dataset/index.html), thereby ensuring a one-to-one mapping between images and instructions. Following the IIT-AFF dataset, the IIT-AFF V-L dataset includes ten object categories and nine affordance classes. The dataset contains 24, 677 affordance regions at the pixel level. The UMD VL dataset comprise of seven affordance classes and 17 object categories. The two datasets are split into training (80%), testing (10%) and validation (10%). The experiments following the dataset split ensuring a comprehensive and fair evaluation for the compared methods.

**Evaluation metric** To assess the performance of our method, we employed two evaluation metrics: structural similarity (denoted as $F_\beta^\omega$) (*Margolin, Zelnik-Manor & Tal, 2014*) and Mean Intersection-over-Union (mIoU). The $F_\beta^\omega$ evaluates the agreement between model's predictions and ground truth and provides insights into the precision in delineating object boundaries. The mIoU calculates the intersection regions over union regions of the predicted segmentation mask and the ground truth, providing a quantitative measure of the accuracy in terms of area overlap.

**Knowledge collection** Addressing the challenge of enabling models to comprehend natural language instructions, particularly in understanding common sense and contextual information, is pivotal in our approach. For instance, interpreting an instruction like 'Use the knife to cut some apple' requires not just recognizing objects in images but also understanding their affordance parts based on common knowledge, a capability that existing methods in affordance detection and visual grounding often lack.

To bridge this gap, we collect object and affordance-related knowledge from three comprehensive knowledge bases: ConceptNet (*Speer, Chin & Havasi, 2017*) is instrumental in providing relational knowledge in a triadic form. This structure is particularly useful for understanding relationships between objects and their affordances. WebChild (*Tandon, De Melo & Weikum, 2017*) offers detailed commonsense knowledge extracted from language usage across major networks. It enriches our model with nuanced insights into everyday object functions and uses. With its vast and diverse content, Wikipedia serves as a rich resource for knowledge. It provides extensive information on a wide range of object types and their associated affordances.

```
{
    "hammer": "A hammer is a tool consisting of a weighted \"head\" fixed to a long handle that is
swung to deliver an impact to a small area of an object. This can be, for example, to drive nails into
wood, to shape metal (as with a forge), or to crush rock. The modern hammer head is typically made
of steel which has been heat treated for hardness, and the handle (also called a haft or helve) is
typically made of wood or plastic.Hammers include sledgehammers, mallets, and ball-peen hammers.
A traditional hand-held hammer consists of a separate head and a handle.",
    "spoon": "A spoon (dipper) is a type of spoon used for soup, stew, or other foods.a typical spoon
has a long handle terminating in a deep bowl.Some spoons involve a point on the side of the basin to
allow for finer stream when pouring the liquid.spoons are usually made of the same stainless steel
alloys as other kitchen utensils; spoon can be made of aluminium, silver, plastics, melamine resin,
wood, bamboo or other materials. the smaller sizes of less than 5 inches in length are used for sauces
or condiments, while extra large sizes of more than 15 inches in length are used for soup or punch.",
    "cup": "cup have handle.handle have cup.A cup has a handle.handle is a part of cup.cup is
related to handle.cup is a synonym of chump. You are likely to find a cup in the cupboard. cup is
related to cup.Beer is served in a cup.A cup is a container.You are likely to find a cup in the
cabinet.Somewhere coffee can be is a cup.cup is related to coffee.a cup is for drinking.a person can
drink from a cup.cup is related to drink.people can fill their cup with coffee.cup is related to coffee
cup.a cup is for filling.a cup is for tea.cup is related to drink.A cup is a type of cup typically used for
drinking hot beverages, such as coffee, hot chocolate, soup, or tea.cups usually have handles and
hold a larger amount of fluid than other types of cup.",
}
```

**Figure 8** **Example of the collected knowledge examples.** The knowledge is stored as key-value pairs, where the entities like 'hammer', 'spoon' and 'cup' are utilized as the keys.

Our collection process involves several steps: (a) We began by gathering facts for each object category. This involved matching the starting or ending nodes in ConceptNet and WebChild with corresponding category labels and Wikipedia articles. (b) To standardize the information, we converted data from ConceptNet and WebChild into sentences, treating each sentence in a Wikipedia article as a standalone fact.(c) We adopted a key-value pair format for the knowledge base, where the key represents an object and the value details facts about the object, including its function, shape, and other relevant attributes. Figure 8 showcases a subset of the knowledge we have accumulated.

In our model, we opted for unstructured knowledge due to its flexibility and ease of updating and expansion, as opposed to the rigid structure of conventional databases. Furthermore, unstructured knowledge more closely mirrors the natural language expression, making it more suitable for our purposes. To ensure the quality and accuracy of the collected knowledge, we conducted manual checks to eliminate biases and redundant information. The final composition of our knowledge base includes contributions from Wikipedia (59.87%), WebChild (27.88%), and ConceptNet (12.25%).

## Quantitative and qualitative results (RQ1)

**Compared method.** In order to validate the effectiveness of our model, we conducted comparisons with eight established methods. These comparisons are detailed in Tables 3 and 4. To ensure a fair and unbiased comparison, all models were trained from scratch with the same sample size and the number of classes. This approach guarantees that any

**Table 3  Performance on the IIT vision language dataset.** The bold text is used to emphasize the highest value for each metric.

| Affordance/Method | Image Only | | | | | | | | Multimodal-based | | | | | |
| --- | --- | --- | --- | --- | --- | --- | --- | --- | --- | --- | --- | --- | --- | --- |
| | DeepLabV 3+ | | OCRNet | | AffordanceNet | GSE | BPN | ADOSMNet | Two-Stage | | BKINet | | KBAG-Net(ours) | |
| | mIoU | $F_\beta^\omega$ | mIoU | $F_\beta^\omega$ | $F_\beta^\omega$ | $F_\beta^\omega$ | $F_\beta^\omega$ | $F_\beta^\omega$ | mIoU | $F_\beta^\omega$ | mIoU | $F_\beta^\omega$ | mIoU | $F_\beta^\omega$ |
| contain | 85.68 | 63.98 | 88.35 | 70.85 | 79.61 | 87.92 | 80.62 | 88.05 | 73.18 | 66.11 | 81.79 | 65.90 | 90.69 | 71.58 |
| cut | 69.29 | 61.58 | 79.20 | 61.94 | 75.68 | 65.34 | 79.23 | 85.63 | 67.45 | 58.68 | 63.37 | 43.01 | 76.38 | 62.73 |
| display | 87.03 | 39.30 | 89.46 | 41.42 | 77.81 | 91.90 | 80.55 | 86.68 | 74.37 | 39.18 | 96.68 | 77.17 | 89.91 | 42.50 |
| engine | 78.6 | 73.99 | 79.82 | 73.96 | 77.50 | 81.91 | 81.49 | 87.01 | 58.94 | 51.53 | 78.52 | 75.87 | 82.82 | 76.91 |
| grasp | 75.95 | 29.28 | 78.59 | 31.61 | 68.48 | 79.76 | 72.96 | 84.99 | 70.60 | 29.42 | 69.12 | 59.31 | 85.92 | 34.64 |
| hit | 85.08 | 40.07 | 94.97 | 39.12 | 70.75 | 90.51 | 88.84 | 86.05 | 89.96 | 35.93 | 64.25 | 75.00 | 93.03 | 40.43 |
| pound | 79.47 | 54.82 | 79.21 | 53.16 | 69.57 | 75.95 | 77.59 | 84.43 | 72.66 | 46.09 | 35.56 | 51.28 | 79.06 | 51.37 |
| support | 79.60 | 67.88 | 81.36 | 61.09 | 69.81 | 78.41 | 80.96 | 84.77 | 76.98 | 56.59 | 71.44 | 68.63 | 86.68 | 64.40 |
| w-grasp | 88.58 | 70.08 | 82.13 | 73.20 | 70.98 | 89.43 | 74.56 | 85.02 | 67.56 | 64.06 | 52.43 | 76.06 | 84.06 | 76.57 |
| Average | 81.03 | 55.66 | 83.68 | 56.26 | 73.35 | 82.33 | 79.64 | 85.85 | 72.41 | 49.73 | 68.13 | 65.83 | **85.39** | 57.90 |

**Table 4  Performance on the UMD vision language dataset.** The bold text is used to highlight the best value for each metric.

| Affordance/Method | Image 0nly | | | | | Multimodal-based | | | | | |
| --- | --- | --- | --- | --- | --- | --- | --- | --- | --- | --- | --- |
| | DeepLabV 3+ | | OCRNet | | AffordanceNet | Two-Stage | | BKINet | | KBAG-Net (ours) | |
| | mIoU | $F_\beta^\omega$ | mIoU | $F_\beta^\omega$ | $F_\beta^\omega$ | mIoU | $F_\beta^\omega$ | mIoU | $F_\beta^\omega$ | mIoU | $F_\beta^\omega$ |
| contain | 85.80 | 81.27 | 85.40 | 81.72 | 59.75 | 86.11 | 83.38 | 79.03 | 47.81 | 85.74 | 82.09 |
| cut | 82.3 | 78.6 | 82.76 | 79.42 | 64.26 | 82.07 | 80.62 | 42.32 | 43.65 | 83.15 | 80.20 |
| grasp | 69.5 | 80.03 | 71.02 | 80.54 | 65.97 | 71.39 | 78.70 | 75.98 | 51.80 | 75.76 | 81.87 |
| pound | 85.42 | 81.29 | 85.61 | 81.70 | 76.73 | 85.20 | 82.08 | 61.45 | 67.17 | 85.32 | 82.91 |
| scoop | 85.57 | 79.74 | 85.92 | 80.65 | 77.0 | 85.79 | 81.65 | 70.84 | 86.83 | 86.38 | 82.79 |
| support | 83.77 | 81.11 | 85.03 | 82.04 | 79.45 | 85.86 | 81.34 | 89.37 | 90.47 | 87.51 | 82.62 |
| w-grasp | 80.80 | 76.04 | 81.01 | 77.06 | 78.65 | 78.97 | 73.00 | 86.46 | 88.08 | 80.38 | 78.67 |
| Average | 81.88 | 79.73 | 82.39 | 80.45 | 71.6 | 82.19 | 80.11 | 72.20 | 67.97 | **83.41** | **81.59** |

observed performance differences are attributable solely to the distinct model architectures and training methodologies.

- DeepLabV3+ (*Chen et al., 2018*) employs dilate convolutions and an encoder–decoder structure to achieve fine-grained segmentation results through multi-scale feature fusion and spatial pyramid pooling.
- OCRNet (*Yuan, Chen & Wang, 2020*) introduces object-contextual representations in conjunction with attention mechanisms. Since affordance segmentation is closely related to object classes, we set OCRNet an appropriate image-only baseline for comparison.
- AffordanceNet (*Do, Nguyen & Reid, 2018*) is a classical affordance detection method. It segments images based on affordance categories without the integration of language instructions, providing a language-agnostic baseline.
- BPN (*Yin & Zhang, 2022*) proposes a boundary-preserving network which considers the relationship between object categories and object affordances. It is an image-only method for affordance detection. Because this dataset is a closed-source dataset, we employed the results of the IIT-AFF dataset reported in the original articles.

- GSE (*Zhang et al., 2022*) utilizes a repeated multi-scale feature-map-fusion network to produce category-relevant feature maps. Because this dataset is a closed-source dataset, we employed the results of the IIT-AFF dataset reported in the original articles.
- ADOSMNet (*Chen et al., 2024*) is the state-of-the-art affordance detection method. We considered it an upper bound in our comparisons since it is an image-only method which eliminates the need to interpret language. We employed the results (with ResNet101) of the IIT-AFF dataset reported in the original articles. Unfortunately, BPN, GSE and ADOSMNet's performance on the UMD VL dataset could not be compared because the UMD vision language dataset delete the repeated images in the UMD dataset to avoid data bias.
- Two-stage method: We design a two-stage approach to integrate Fast R-CNN (*Ren et al., 2015*) and DeepLabV3+ as a baseline method. Firstly, Fast R-CNN is used to detect and crop the target objects. Subsequently, the cropped regions are input into DeepLabV3+ to generate affordance segmentation masks for the targets. This approach can provide deep insights into the impact of language composition in the affordance grounding task.
- BKINet (*Ding et al., 2023*) represents the state-of-the-art multi-modal method with knowledge incorporation. It is designed for the referring image segmentation task, incorporating specific knowledge of the target object in the image.

The image-only methods provide an access to the impact of language instructions. The two-stage method and multi-modal methods evaluate how different integration of text and image data affect the performance for affordance grounding. Different kinds of methods provide comprehensively evaluate of our method for language-following affordance segmentation and multimodal understanding.

**Quantitative evaluation.** Table 3 summarizes the results of the IIT-AFF VL dataset. The results clearly show that KBAG-Net has achieved substantial improvements. Compared to DeepLabV3+, our approach exhibits a notable increase in performance, demonstrating a gain of +2.9% in mIoU and +2.35% in $F_{\beta}^{\omega}$, respectively. Against OCRNet, our method shows superior results with increases of +1.72% in mIoU and +1.64% in $F_{\beta}^{\omega}$. It is noteworthy that KBAG-Net achieves these results using an end-to-end architecture, and no further post-processing step like object detection and crop as the two-stage method. Our approach displays a significant enhancement over two-stage method with improvements of +12.98% in mIoU and +8.17% in $F_{\beta}^{\omega}$. While our $F_{\beta}^{\omega}$ scores are slightly lower than those of KBAG-Net, our model surpasses KBAG-Net in mIoU by a significant margin of +17.26%. The higher mIoU with lower F-score could be due to imbalanced class distributions. The discrepancy of $F_{\beta}^{\omega}$ scores may be attributed to our method's handling of certain affordance classes, notably "grasp" and "hit". These classes often present significant challenges in accurately delineating edge regions, which is crucial for achieving high $F_{\beta}^{\omega}$ scores. Therefore, the model has high precision but low recall for certain affordance classes, leading to a lower average F-score. Overall, our method has a stronger performance in terms of overall segmentation accuracy.

Table 4 provides a comprehensive summary of our model's performance on the UMD VL dataset. In these experiments, KBAG-Net consistently outperforms other comparative

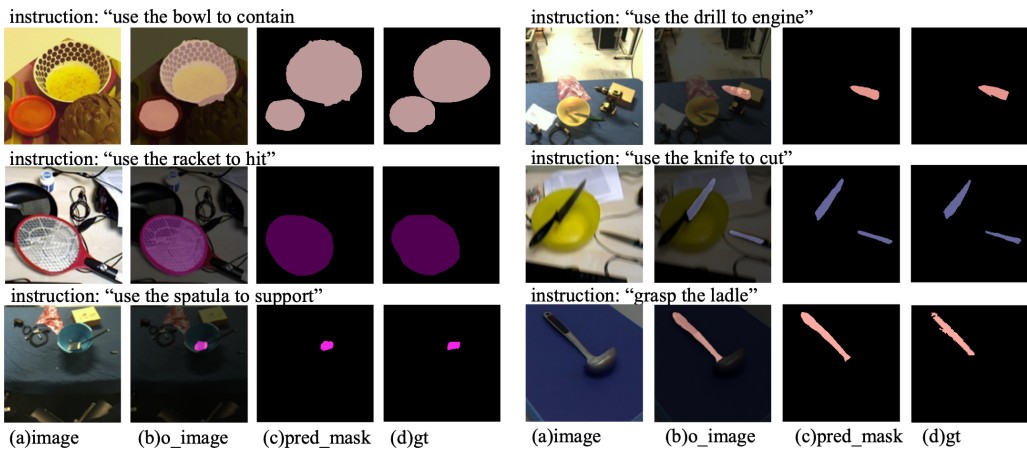

**Figure 9  Visualization of the final results by our model.** 'o_image' is the overlay of the predicted mask on the original image, 'pred_mask' is the predicted mask, and 'gt' represents the ground truth segmentation mask of the image. Image source credits: *Nguyen et al. (2017)* and *Myers et al. (2015)*.

methods, affirming its robustness and effectiveness. Specifically, our model surpasses the second best (OCRNet) +1.02% in mIoU and +1.14% $F_\beta^\omega$. It is worth noting that the UMD dataset only contains clutter-free scenes. Therefore, the improvement of KBAG-Net over compared methods is not as significant as those observed on the IIT-AFF dataset.

In conclusion, the results from our extensive experiments, as detailed in Table 3 and Table 4, clearly demonstrate that our KBAG-Net method marks a significant advancement over existing technologies in the realm of fine-grained part-level affordance segmentation. A key strength of our approach is its ability to operate in an end-to-end manner, effectively eliminating the need for any additional post-processing or data augmentation steps. This streamlined process not only simplifies the workflow but also contributes to the robustness of the results. The performance across different affordance categories, as reflected in the mIoU and $F_\beta^\omega$ scores, reveals insightful trends. The ''grasp'' category presents a more challenging scenario. This is primarily due to the complexity of scenes and the irregular shapes of objects requiring grasping, which inherently lead to greater difficulties in achieving precise segmentation.

**Qualitative results.** To further validate the effectiveness of our proposed module in identifying the object's graspable region and generating accurate mask predictions, we showcase a selection of results in Fig. 9. These visual representations show that our method can effectively detect part-level regions of masks according to the language, and the predicted results are close to the ground truth.

## Ablation studies (RQ2)

In order to comprehensively understand the contribution of each component within our framework, we conducted a series of ablation studies. These studies are essential for isolating and evaluating the impact of individual elements of our model on its overall performance.

**Table 5  Ablation study on the IIT-AFF vision language dataset.** The symbol ✓ indicates the addition of the corresponding component. Bold text is utilized to emphasize the best values achieved for each metric.

| | Knowledge enhancement | | Multimodal fusion | | Metrics | | | | |
|---|---|---|---|---|---|---|---|---|---|
| | w/KB | w/BLFI | w/BULG | Attention | Overall Acc | Mean Acc | FreqW Acc | Mean IoU | $F_\beta^\omega$ |
| 1 | – | – | – | – | 95.13 | 81.33 | 90.93 | 71.89 | 52.01 |
| 2 | ✓ | – | – | – | 95.87 | 82.83 | 92.19 | 73.43 | 52.38 |
| 3 | ✓ | ✓ | – | – | 96.26 | 81.78 | 92.93 | 73.49 | 52.88 |
| 4 | – | – | ✓ | – | **96.36** | 81.79 | **93.13** | 74.22 | 53.33 |
| 5 | ✓ | – | ✓ | – | 96.01 | 84.15 | 92.46 | 76.16 | 53.35 |
| 6 | ✓ | ✓ | – | a | 95.87 | 83.67 | 92.28 | 73.45 | 51.98 |
| 7 | ✓ | ✓ | – | b | 95.25 | 86.98 | 91.34 | 72.67 | 51.50 |
| 8 | ✓ | ✓ | – | c | 95.75 | 82.56 | 91.98 | 75.16 | 52.39 |
| 9 | ✓ | ✓ | ✓ | – | 96.21 | **85.83** | 92.80 | **78.40** | **54.00** |

**The components of the network.** In validate the effectiveness of the BULG module, Knowledge, and BLFI module, we conducted ablation studies on IIT-AFF VL dataset, with the results presented in Table 5. The notation 'w/KB' indicates the incorporation of the knowledge information. The notations 'w/BLFI' and 'w/BULG' denote the addition of the Bimodal Language Feature Interaction and Bottom-up Language-guided Multi-modal Fusion, respectively. The 'Attention' column refers to the three types of typical multimodal attention mechanism displayed in Fig. 5. The baseline method (first row of Table 5) combines image features with text features *via* simple concatenation. The ablation studies reveal significant findings:

(1) Knowledge based only (second row of Table 5): The introduction of external knowledge alone enhances the model's understanding of natural language instructions, evidenced by increases in mIoU and $F_\beta^\omega$ by 1.24% and 0.37%, respectively, compared to the baseline.

(2) Knowledge with BLFI module (fourth row of Table 5): The incorporation of the BLFI module to filtering text information noise results in further improvements in text features, with subsequent increases in mIoU and $F_\beta^\omega$.

(3) Knowledge with BULG module (fifth row of Table 5): The fusion of knowledge information with textual feature, combined with the BULG module's dense multimodal information interaction, significantly aligns visual and text features. This configuration shows a substantial improvement over the baseline, with mIoU and $F_\beta^\omega$ increasing by 2.33% and 1.32%, respectively.

(4) Full integration (last line of Table 5): Utilizing all three components simultaneously results in the most significant performance enhancements, with increase of 6.51% in mIoU and 1.99% in $F_\beta^\omega$ compared to the baseline.

Further evaluations were conducted to assess different attention mechanisms for multimodal fusion, with results from the sixth to ninth rows of the Table 4 illustrating the effectiveness of replacing the attention mechanism in multimodal fusion with three alternative mechanisms as shown in Fig. 5. The outcomes confirm the effectiveness of the proposed BULG module.

| Table 6   Model efficiency analysis. | | | | |
|---|---|---|---|---|
| **Model** | **FPS** | **Params** | **mIoU** | $F_\beta^\omega$ |
| DeepLabV 3+ | 4.28 | 39.75 | 82.49 | 55.55 |
| Two-Stage | 2.61 | 81.86 | 72.41 | 49.73 |
| BKINet | 10.85 | 15648 | 68.13 | 65.83 |
| KBAG-Net | 6.16 | 68.26 | 85.39 | 57.90 |

**Model efficiency analysis.** As shown in Table 6, we calculate the number of parameters and FPS for comparison methods. Our method achieves the best performance in terms of mIoU while utilizing fewer parameters compared to two other multimodal methods. Compared to DeepLabV3+, our model has demonstrated a significant improvement in terms of computational overhead, processing speed, and performance, with enhancements of +2.9% and +2.35% on IIT-AFF VL dataset, respectively. Compared to the two-stage methods, we achieved better performance and processing speed at a relatively lower computational overhead, resulting in increases of +12.98% and +8.17% on IIT-AFF. Although our computational overhead and processing speed do not match the levels of BKINet, our performance has shown improvements of +17.26% on the mIoU metric. The total number of parameters in our CNN model is less than half that of the BKINet.

## Data quality and efficiency analysis (RQ3 and RQ4)

To evaluate the proposed semi-automatic method whether generate high-quality dataset, we used the framework to generate two datasets, which is used to validate the quality. We conducted experiments on two publicly available datasets: IIT-AFF dataset (*Nguyen et al., 2017*) and UMD dataset (*Myers et al., 2015*), which have ground truths of object detection and affordance segmentation. We add manipulation instructions for part of the dataset manually, and compare the generated annotations (instructions, object bounding boxes and affordance masks) with the ground-truth to evaluate the quality of the generated data by our method.

**Generated data analysis.** Some examples of generated data are shown in Fig. 10. The left examples are generated from the IIT-AFF dataset and the right examples are generated from the UMD dataset. For the instructions, we mixed the generated instructions and manual annotations together, and selected five volunteers to choose the generated instructions from the mixed instructions. The test shows that it is difficult to distinguish the generated instructions from the manual annotations. For the quality of object bounding boxes, we evaluate the performance based on the object detection mean average precision (mAP). The mAP achieved on the IIT-AFF dataset is 80%, and that on the UMD dataset is 99%. The results show that our method can generate both object-level and part-level annotation for implicit and explicit instructions automatically, which improves the data annotation efficiency for the manipulation grounding task. The false annotation in the generated data could be further removed by humans, which is much easier than data labeling.

**Data construction efficiency analysis.** For the IIT-AFF and UMD datasets, we generated two types of instructions for each object class. Table 7 presents the statistical information of the vision-language dataset generated based on the IIT-AFF and the UMD datasets.

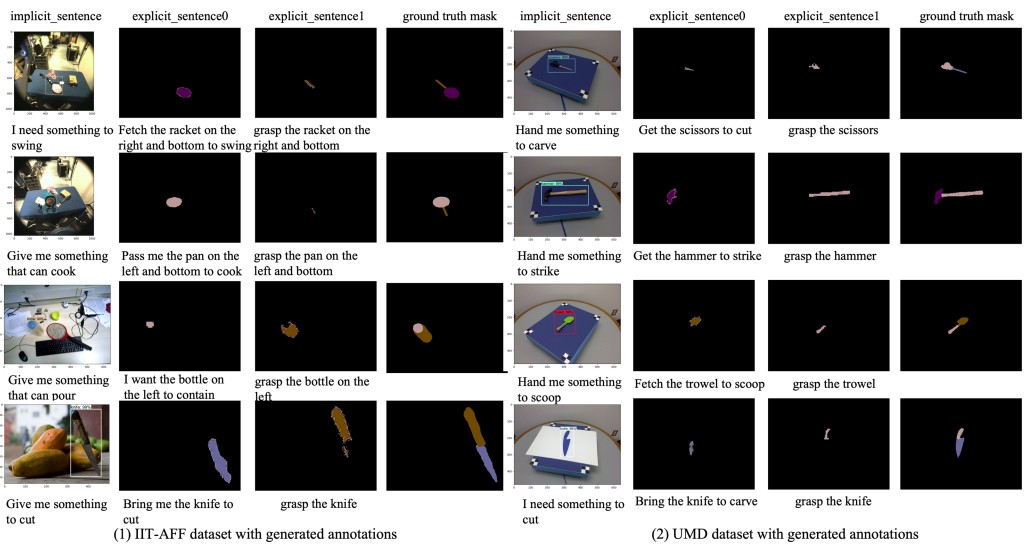

**Figure 10   Visualization of the IIT-AFF dataset and the UMD dataset with generated instructions and image annotations.** Image source credits: *Nguyen et al. (2017)* and *Myers et al. (2015)*.

We list the explicit and implicit instructions for the corresponding object classes. To generate the object bounding boxes and the instructions, it takes approximately 75 hours for the two datasets. The affordance masks generation takes approximately 9 and a half hours. According to prior study *Papadopoulos et al. (2017)*, skilled annotators may need an average of 79s to label a polygon-based instance mask for an image in MS COCO. Compared with human annotation, our method could save considerable time and reduce manual annotation efforts for dataset construction.

**Organization and application of the datasets.** We use a structured JSON file format to store these data, ensuring that they are easily accessible and well-organized for various applications. Each entry in the JSON file corresponds to an image and contains a set of key properties related to the detected objects in that image. These properties include box, explicit_sentence, guid, implicit_sentence, label. 'guid' represents the object id number for each annotation. The specific format of this JSON structure is illustrated in Fig. 11, providing a visual reference for clarity.

Additionally, we employ a naming convention for object part-level masks to ensure consistency and easy retrieval. These mask filenames follow a specific format that includes the RGB name, index references, and object-specific information. For example, "ILSVRC2014_train_00059832_0_0.png" has "ILSVRC2014_train_00059832" as the RGB name, "first-index" corresponds to the index of the value array in the JSON file corresponding to the image name, and "second-index" represents the index of the object's explicit instruction array. This structured organization and naming convention enhance the usability of our dataset, making it well-suited for a wide range of robotic manipulation and language-grounding tasks.

Using object visual grounding as a test, we evaluate the performance of the TransVG (*Deng et al., 2021*), which is an end-to-end visual grounding model, on the IIT-AFF and

**Table 7  Automatic generated data statistics of instruction for IIT-AFF and UMD dataset.**

| dataset | category | explicit_instruction | implicit_instruction |
|---------|----------|----------------------|----------------------|
| IIT-AFF | bowl | 845 | 513 |
| | tvm | 2,306 | 2,750 |
| | pan | 2,555 | 1,707 |
| | hammer | 3,152 | 1,943 |
| | knife | 2,632 | 1,826 |
| | cup | 2,957 | 1,942 |
| | racket | 2,475 | 1,620 |
| | bottle | 2,482 | 2,120 |
| UMD | bowl | 1,855 | 1,855 |
| | cup | 2,792 | 1,396 |
| | hammer | 2,232 | 1,116 |
| | knife | 6,722 | 3,449 |
| | ladle | 2,426 | 1,346 |
| | mug | 8,882 | 5,249 |
| | saw | 1,607 | 804 |
| | scissors | 3,516 | 2,026 |
| | scoop | 1,156 | 595 |
| | shears | 789 | 467 |
| | spoon | 5,604 | 2,871 |
| | trowel | 240 | 120 |

UMD datasets with generated instruction(Using ground-truth image annotation). The TransVG predicts the object bounding box according to the image and language. As shown in Table 8, we assess its accuracy in two scenarios: using explicit and implicit instruction as language expressions for visual grounding. The accuracy percentages refer to the localization accuracy of object bounding boxes. For the TransVG model, we utilize both ResNet50 and ResNet101 as convolutional backbone networks. Under the ResNet-101 backbone, TransVG achieves accuracy rates of 88.00% and 99.86% on the generated IIT-AFF and UMD datasets based on explicit instructions, respectively. However, when it comes to implicit instruction, TransVG achieves 75.14% and 99.76% with the same backbone. The high accuracy achieved on the UMD dataset can be attributed to the dataset's unique characteristics, where each image contains only one object, captured from various angles, making object localization relatively straightforward.

The results illustrate that the difficulty of the benchmark dataset is determined by at least two aspects: *the diversity of the objects appearance* and *the complexity of the language instructions*. Since the object in the UMD dataset is lack of variation, the complexity of instructions has limited influence for the performance(0.11 and 0.1 for two backbones). In contrast, the images in the IIT-AFF dataset have more diversity in the object appearance, the performance drop about 12.32 and 12.86 for the implicit instructions with two different backbones. Compared with existing datasets of visual grounding, our generated language-grounded manipulation dataset (IIT-AFF implicit) demonstrates a level of difficulty similar to that of Flickr30K Entities (*Plummer et al., 2015*) and presents a more challenging task

```json
{
        "ILSVRC2014_train_00059832.jpg":[
            {
              "box" :[
                93,
                12,
                197,
                480
              ],
              "explicit_sentence" :[
                  "I want the bottle on the left to contain",
                  "Grasp the bottle on the left"
              ]
              "guid" : 0,
              "implicit_sentence": "Give me an item that can contain",
              "label" :"bottle"
            },
            {
              "box":[
                190,
                10,
                307,
                461
              ],
              "explicit_sentence":[
                "Fetch the bottle on the right to contain",
                "Grasp the bottle on the right"
              ],
              "guid" :1,
              "implicit_sentence": "Give me something that can pour",
              "label" :"bottle"
            }
        ]
}
```

**Figure 11  JSON format of the generated dataset.**

than RefCOCO (*Yu et al., 2016*). Besides the REC task, the generated dataset can also be applied to the RIS task. In future work, we plan to assess the suitability of existing RIS methods on our generated language following manipulation datasets. Additionally, we anticipate that our dataset may have broader applications beyond the REC and RIS tasks, which we will explore in subsequent research.

**Table 8** Performance of the TransVG model on the language following IIT-AFF and UMD datasets, as well as three visual grounding datasets. Accuracy (%) is used as the metric for performance.

| dataset | backbone | Accuracy |
|---|---|---|
| IIT-AFF(explicit) | ResNet50 | 87.28 |
| | ResNet101 | 88.00 |
| IIT-AFF(implicit) | ResNet50 | 74.96 |
| | ResNet101 | 75.14 |
| UMD(explicit) | ResNet50 | 99.78 |
| | ResNet101 | 99.86 |
| UMD(implicit) | ResNet50 | 99.67 |
| | ResNet101 | 99.76 |
| ReferItGame | ResNet50 | 69.76 |
| | ResNet101 | 70.73 |
| Flickr30K | ResNet50 | 78.47 |
| | ResNet101 | 79.10 |
| RefCOCO | ResNet50 | 82.67 |
| | ResNet101 | 82.72 |

## CONCLUSIONS

In this article, we present a novel knowledge enhanced bottom-up affordance grounding network that is designed for fine-grained part-level affordance segmentation. One of the standout contributions is the integration of knowledge through the BLFI module, demonstrating the significant impact of tailored knowledge in the context of affordance segmentation. Additionally, the introduction of the BULG module showcases how the fusion of visual features, guided by enhanced text features, can substantially improve the learning of correlations between image regions and language instructions. Finally, we introduce a unified framework for semi-automatic dataset generation. The pipeline showcases the ability to automatically generate instructions with multiple types and annotate images with multiple granularities. Experimental results affirm the effectiveness of incorporating external knowledge in the comprehension of natural language commands. The unified framework significantly reduces the manual effort involved in dataset construction for diverse application scenarios. The research of affordance understanding and grounding holds significant promise for enhancing robot interaction capabilities in various domains. By imbuing robots with the ability to perceive and understand affordances in their environment, they can better interpret and respond to human intentions, leading to more intuitive and efficient human–robot interactions. In future, understanding implicit language instruction poses a set of challenges within existing visual grounding methods, demanding further exploration.

### Funding

The authors received no funding for this work.

## Competing Interests

The authors declare there are no competing interests.

## Author Contributions

- Wen Qu conceived and designed the experiments, performed the experiments, analyzed the data, performed the computation work, prepared figures and/or tables, authored or reviewed drafts of the article, and approved the final draft.
- Xiao Li performed the experiments, analyzed the data, performed the computation work, prepared figures and/or tables, authored or reviewed drafts of the article, and approved the final draft.
- Xiao Jin performed the experiments, analyzed the data, performed the computation work, prepared figures and/or tables, authored or reviewed drafts of the article, and approved the final draft.

## Data Availability

The IIT-AFF VL and UMD VL datasets are available in the Supplemental Files. They are available at:

- https://sites.google.com/site/iitaffdataset
- https://users.umiacs.umd.edu/~fer/affordance/part-affordance-dataset/index.html.

## Supplemental Information

Supplemental information for this article can be found online at http://dx.doi.org/10.7717/peerj-cs.2097#supplemental-information.

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
