# Peer review of "Knowledge enhanced bottom-up affordance grounding for robotic interaction"

_PeerJ Computer Science, doi:10.7717/peerj-cs.2097_

## Round 0.1 · original submission · Major Revisions

Please carefully made changes to your manuscript according to the reviewers' comments for further review.

Reviewer 1 ·

Basic reporting

This paper introduces a novel fine-grained affordance prediction framework and a semi-automated data annotation method, which are both beneficial to the field. However, there are several issues that require revision and clarification:

1. The term "knowledge enhanced bottom-up language-guided affordance grounding network (KBAG-Net)" is quite lengthy and difficult to comprehend. The description between lines 60 to 61 only covers "knowledge enhanced" and should be complemented with explanations for "bottom-up" and "language-guided" aspects.

2. It is not advisable to use the word "most" without data support (line 74).

3. There is a misstatement in lines 130 to 131, as the subject responsible for "extracting visual features" is not the "language instruction".

4. The organization of the method section is confusing. It is customary to introduce the construction of the dataset before the method, but in this paper, the method framework is sandwiched between two parts of data collection (Knowledge Collection at line 136 and SEMI-AUTOMATIC DATASET CONSTRUCTION at line 245).

5. There is a discrepancy between the text and the figure in lines 226 to 232, such as the reference to "the left part" in line 227, which should actually refer to the right side of Figure 7.

Experimental design

The experimental section is well-organized, but the conclusions would be clearer if the following questions are addressed:

1. The meaning expressed in lines 350 to 351 seems to suggest that the dataset used for training and evaluating the method in this paper is not the one produced through the semi-automated process in this paper. If this is not the case, the differences between the two datasets need to be clarified, along with the specific role of the data produced through the semi-automated process.

2. Figure 9 appears between lines 351 and 352 without any explanation or introduction. A natural assumption would be that it presents a dataset from the prior work (https://github.com/WenQu-NEU/Affordance-Grounding-Dataset). However, based on the description at line 476, it appears to be the dataset generated by the semi-automated annotation process in this paper. Better organization and clearer clarification are needed here.

3. The changes made to the BULG module in the ablation study are not clearly described in lines 450 to 459. My understanding is that BULG uses both low-level and high-level visual features, as shown in Figure 3. What do the fourth and fifth rows in Table 4 specifically represent? My guess for the fourth row is: ① using only high-level features and eliminating low-level features; ② using high-level features in two channels of BULG. The experiment results of two guess above need to be presented and clearly explained.

4. As shown in Figure 6, the feature fusion in this paper is a key innovative point. Adding comparative results with modules (a)-(c) would better support the design theory.

Validity of the findings

no comment

Cite this review as

·

Basic reporting

Summary

The manuscript is well organized and written, easy to understand, and provides detailed background on the field. This manuscript proposes a novel knowledge-enhanced bottom-up affordance grounding network (KBAG-Net), which utilizes external knowledge to enhance natural language understanding. This improvement in understanding leads to increased accuracy in object grasping and affordance segmentation. Additionally, they introduce a semi-automatic data generation method designed to expedite the creation of a language-guided manipulation grounding dataset.

Experimental design

The manuscript compares eight existing models to demonstrate the effectiveness of the proposed KBAG-Net and presents detailed ablation experiments that illustrate the rationale and effectiveness of the designed module. Besides, the manuscript releases semi-automatic data, which is meaningful for promoting robotics research.

Validity of the findings

The manuscript's well-designed experiments address the raised questions. I think the author's proposed method of integrating knowledge is instructive for resolving the hallucination problem in current large language models.

Additional comments

Weakness

1. Gap Analysis:Strengthen the discussion on the limitations of existing approaches and how your method addresses these gaps. This will further justify the need for your research.

2. In Table 3, the performance differences between different models are obvious, and the author needs to provide further explanation.

3. In Table 5, what's the difference between the fourth and fifth lines? In line 498,what's the meaning of 'guid'.

4. Consistent punctuation should be added to all formulas.

Reviewer 3 ·

Basic reporting

This research introduces a novel knowledge-enhanced bottom-up affor part-level affordance grounding in response to natural language commands and presents the KBAG-Net, a multimodal attention-based model that integrates language and vision features for direct affordance part identification. By leveraging advanced feature fusion strategies, the model achieves superior accuracy and efficiency over existing methods, with significant improvements demonstrated on two vision-language affordance datasets.

Experimental design

The manuscript demonstrates technical correctness, with a solid foundation in its approach to integrating external knowledge for affordance segmentation. The methodology was sound and the experiments results were well-presented, showcasing a rigorous scientific analysis that supports the claim made. The proposed semi-automatic data construction method was thoroughly presented.

Validity of the findings

This paper addresses the problem of affordance grounding and proposes a useful framework for dataset semi-automatic contruction. This technique is very effective compared with standard techniques in terms of combining image and knowledge enhanced language information. Cosider the incorporation of the application of your method to other domains to broaden the scope of this paper, such as operation guide in education and operation intent prediction.

Additional comments

Introduction and related work are quite well written.
Line 197, K_2 and V_2 ->K_1 and V_1?
In equation (3), mearning of d_k was not detailed.
Line 238, please better define f_m1.
Line 281, the authors should correct the equation citation.

Cite this review as

---

## Round 0.2 · accepted · Accept

Based on the reviewers' comments, the manuscript may now be accepted.

Reviewer 1 ·

Basic reporting

This paper introduces a novel fine-grained affordance prediction framework and a semi-automated data annotation method, which are both beneficial to the field. The authors have responded to the questions raised and there are no further comments.

Experimental design

The experimental section is well-organized. The authors have responded to the questions raised and there are no further comments.

Validity of the findings

no comment

Cite this review as

Reviewer 3 ·

Basic reporting

The paper is well-organized, and the writing is clear and concise, making the complex information accessible and engaging for readers. All the comments have been well addressed.

Experimental design

The methodology is well-designed and appropriately executed. The choice of each component is suited to the research questions posed.

Validity of the findings

The analysis is thorough, and the conclusions drawn are supported by the data. The discussion integrates the findings with current knowledge in the field, enhancing the paper's contribution.

Cite this review as